# The Influence of Prone Positioning on Energy and Protein Delivery in COVID-19 Patients Requiring ECMO Support

**DOI:** 10.3390/nu16203534

**Published:** 2024-10-18

**Authors:** Marlene Hintersteininger, Patrick Haselwanter, Mathias Maleczek, Daniel Laxar, Martina Hermann, Alexander Hermann, Nina Buchtele, Thomas Staudinger, Christian Zauner, Mathias Schneeweiss-Gleixner

**Affiliations:** 1Department of Medicine III, Clinical Division of Gastroenterology and Hepatology, Medical University of Vienna, 1090 Vienna, Austria; marli.hintersteininger@gmail.com (M.H.); patrick.haselwanter@meduniwien.ac.at (P.H.); christian.zauner@meduniwien.ac.at (C.Z.); 2Department of Anesthesia, Intensive Care Medicine and Pain Medicine, Medical University of Vienna, 1090 Vienna, Austria; mathias.maleczek@meduniwien.ac.at (M.M.); daniel.laxar@dhps.lbg.ac.at (D.L.); martina.hermann@meduniwien.ac.at (M.H.); 3Ludwig Boltzmann Institute for Digital Health and Patient Safety, Medical University of Vienna, 1090 Vienna, Austria; 4Department of Medicine I, Intensive Care Unit 13i2, Medical University of Vienna, 1090 Vienna, Austria; alexander.hermann@meduniwien.ac.at (A.H.); nina.buchtele@meduniwien.ac.at (N.B.); thomas.staudinger@meduniwien.ac.at (T.S.)

**Keywords:** critical care, medical nutrition therapy, prone position, ECMO, COVID-19

## Abstract

Background: Gastrointestinal dysfunction is a common complication of medical nutrition therapy in critically ill patients. Whether prone positioning leads to a deterioration in gastrointestinal function has not been fully clarified. Thus, we aimed to analyze the influence of prone positioning on the tolerance of medical nutrition therapy. Methods: We conducted a retrospective analysis of 102 SARS-CoV-2 infected patients with venovenous extracorporeal membrane oxygenation support (VV ECMO). Gastric residual volume (GRV) was used to assess the tolerance of enteral nutrition. Results: Nutritional data were collected for 2344 days. Undernutrition was observed in 40.8%, with a significantly higher incidence on days in prone position (48.4% versus 38.6%, *p* < 0.001). On days in supine position, significantly more calories were administered enterally than on days in prone position (*p* < 0.001). The mean GRV/24 h was 111.1 mL on days in supine position and 187.3 mL on days in prone position (*p* < 0.001). Prone positioning was associated with higher rates of GRV of ≥500 mL/24 h independent of age, disease severity at ECMO start, ECMO runtime and ICU length of stay (adjusted hazard ratio: 4.06; 95%CI: 3.0–5.5; *p* < 0.001). Conclusions: Prone position was associated with lower tolerance of enteral nutrition, as indicated by an increased GRV. As a result, reduced enteral nutritional support was administered.

## 1. Introduction

The Coronavirus Disease 2019 (COVID-19) pandemic resulted in a dramatic increase in acute respiratory distress syndrome requiring intensive care treatment. In addition to standard medical procedures like invasive mechanical ventilation and prone positioning, venovenous extracorporeal membrane oxygenation (VV ECMO) was required in a significant proportion of COVID-19 patients [1,2]. Due to increased protein metabolism caused by the inflammatory reaction of the acute illness and the ECMO therapy itself, hemodynamic instability, impaired microcirculation and gastrointestinal (GI) dysfunction, adequate nutrition support represents a clinical challenge in this patient population [3,4,5,6,7]. Therefore, ECMO patients are at high risk of developing malnutrition, ultimately leading to increased morbidity and mortality [2,8,9,10]. Based on these data, the diagnosis and avoidance of malnutrition should be of great importance in the treatment of ARDS patients requiring ECMO support [3,11].

GI dysfunction is a common complication of medical nutrition therapy in critically ill patients. GI dysfunction is associated with patient morbidity, life-threatening emergencies and worse clinical outcomes [12]. However, a standardized definition of GI dysfunction, adequate monitoring and evidence-based therapeutic approaches are still missing [12,13]. The role of gastric residual volume (GRV) measurements for the diagnosis of GI dysfunction remains controversial [14,15,16]. Nevertheless, the GRV measurement is recommended in current guidelines to assess gastrointestinal dysfunction [17].

Despite the aforementioned challenges and barriers for adequate nutrition support, there are currently no specific nutrition guidelines either for patients undergoing ECMO therapy or for patients infected with COVID-19 regardless of ECMO support [18,19,20,21]. Most recommendations are based on existing nutrition guidelines [17,22]. In addition, the influence of prone positioning on the efficacy and tolerance of medical nutrition support, especially enteral nutrition (EN), has still not been fully clarified [23]. Nevertheless, according to recent guidelines, the prone position itself represents no contraindication for enteral feeding [17,19]. Indeed, there is evidence that the gastrointestinal tolerance represented by GRV does not differ when comparing supine and prone positions in mechanically ventilated patients [23,24,25]. However, there are no existing data on the influence of prone positioning on the tolerance of enteral nutrition support in ECMO patients. Based on the available data, we hypothesized that prone positioning has no effect on the tolerance of medical nutrition therapy in ECMO patients.

Thus, this study aimed to (i) analyze the nutrition support practice according to current guidelines of the “European Society for Clinical Nutrition and Metabolism” (ESPEN) on clinical nutrition in the ICU [17] and (ii) to evaluate the influence of prone positioning on the feasibility and tolerance of medical nutrition therapy in a large single tertiary center cohort of COVID-19 patients on VV ECMO support.

## 2. Patients and Methods

### 2.1. Patient Cohorts

This study is a subanalysis of the previously published study of Schneeweiss-Gleixner et al. [26]. We conducted a retrospective observational study of the same 102 COVID-19 patients receiving VV ECMO support between March 2020 and May 2021 at a large tertiary center in Vienna (Medical University of Vienna, Austria). During the first three surges of the COVID-19 pandemic, a total of six ICUs (medical and surgical) were responsible for the care of patients with COVID-19-related acute respiratory distress syndrome (ARDS). To investigate the patients’ nutritional support during VV ECMO therapy, the observation period was defined from the first day of VV ECMO support until the day before ECMO decannulation or death during ongoing VV ECMO therapy [26].

This study was approved by the local ethics committee of the Medical University of Vienna (ethic vote number: 2440/2020) and was conducted in accordance with the latest version of the Declaration of Helsinki.

### 2.2. Data Collection

Data collection for each patient referred to the days on which VV ECMO therapy was performed in the ICU. Data were extracted from the patient data management system (IntelliSpace Critical Care and Anesthesia, Philips, Amsterdam, The Netherlands) used in all ICUs at the Medical University of Vienna. The severity of critical illness and the extent of organ dysfunction were calculated using the simplified acute physiology score II (SAPS II) and the sequential organ failure assessment (SOFA) score [27,28].

Prone positioning was performed according to current guidelines with an attempt to maintain it for at least 16 h [29]. Information about the position (supine vs. prone) was collected for each day on ECMO support. The positioning was categorized as prone whenever the patient had been in prone position for at least one hour that day.

### 2.3. Nutrition Related Data

All patients received medical nutrition therapy according to standard ICU procedures. The aim was to start nutrition support via a nasogastric tube within 24 h after admission to the ICU. Enteral nutrition was preferred over parenteral nutrition whenever possible. The type of nutrition support and way of administration (enteral [EN], total parenteral [tPN], supplemental parenteral nutrition [sPN], gastric tube or post-pyloric tube) was analyzed for each day [26].

Data on nutrition intake (i.e., energy and protein from enteral and parenteral nutrition as well as propofol) was collected for each day on VV ECMO support. The energy target was defined according to current ESPEN guidelines using weight-based equations [17]. Energy targets were calculated with 25 kcal per kg actual body weight per day (kcal/kg BW/d) for each day of VV ECMO support. For obese patients (i.e., BMI ≥ 30 kg/m^2^), the adjusted body weight was used according to ESPEN guidelines [17]. Protein delivery per 24 h was defined as g/kg of actual body weight (g/kg BW) for every patient. Adequate nutrient intake was defined as 70–100% of the calculated energy requirements for an entire day, with undernutrition defined as <70% and overnutrition as >100% [17].

In order to assess the tolerability of enteral nutrition in prone position, the GRV per 24 h was determined for each day with VV ECMO support. The cut-off for a high GRV was set at 500 mL/24 h. In addition, the frequency of prokinetic therapy as well as metabolic complications (hypertriglyceridemia = triglycerides ≥ 350 mg/dL; hyperglycemia = glucose ≥ 200 mg/dL) were assessed for each day on ECMO support [26].

### 2.4. Statistical Analysis

All statistical analyses were performed using IBM SPSS Statistics 27 (IBM, New York, NY, USA) and GraphPad Prism 8 (GraphPad Software, San Diego, CA, USA). To identify differences in baseline characteristics, Pearson’s chi-square or Fisher’s exact tests were used to compare categorical variables as appropriate. For nominal and ordinal parameters, absolute numbers with relative frequencies were calculated. Metric variables were tested for normal distribution using the Kolmogorov–Smirnov test. If confirmed, the mean and standard deviation were determined. The t-test was used to compare normally distributed metric variables for statistical significance. For non-normally distributed metric variables, the median and interquartile range (IQR) were calculated and tested for statistical significance using the Mann–Whitney U or the Wilcoxon signed-rank test. The significance level was set at α = 5%, so that a *p* < 0.05 was considered as significant in the hypothesis test.

The primary outcome was defined as the feasibility and tolerance of energy (specified as the percentage of calculated requirements) and protein intake (specified as g/kg BW/d) according to ESPEN guidelines in different positions (supine vs. prone) [17]. The following calculations were performed: First, we conducted the overall daily energy and protein intake of all patients in different positions during the entire observation period. Since the patients were turned to prone position particularly at the beginning of their ECMO support, we further analyzed the mean calorie and protein intake during the first 30 days of ECMO support across the following 4 time periods: the first 3 days, the first week (days 1–7), the second week (days 8–14) and days 15–30 of ECMO support. For further differentiation of the medical nutrition therapy, a subgroup analysis was performed according to the adequacy of nutrition.

The GRV was used to assess the position-dependent tolerance of enteral nutrition. Several calculations were carried out for this purpose:-Daily GRV = mean ± standard deviation of the GRV (mL/24 h) of all patients during the entire observation period;-Time course of daily GRV = mean value ± standard deviation of the GRV (mL/24 h) in the time periods described above;-Limit of the GRV according to ESPEN guidelines [17] = absolute and relative frequency of days with high GRV (i.e., GRV ≥ 500 mL/24 h) during the entire observation period;-Univariable and multivariable Cox regression analyses were used to evaluate parameters associated with a high GRV (i.e., GRV ≥ 500 mL/24 h).

Finally, we evaluated the patient-specific tolerance of enteral nutrition in different positions. For this purpose, we divided the patient cohort into two subgroups: (A) patients who spent all the days of ECMO therapy in supine position and (B) patients who alternated between supine and prone position during their ECMO support. One patient spent his whole ECMO runtime in prone position and was therefore excluded from this calculation. To evaluate the patient-specific tolerance of enteral nutrition in supine and prone position, we compared the mean GRV in supine versus prone position for each patient in subgroup B.

## 3. Results

### 3.1. Baseline Characteristics

A detailed description of the baseline characteristics can be found in the study of Schneeweiss-Gleixner et al. [26] and in Table 1. In the observation period from March 2020 to May 2021, a total of 102 patients with VV ECMO support due to COVID-19-induced ARDS were included. The majority of the patients were male (71.6%). Upon ICU admission, the median age was 57 years (50–62 years) with a median BMI of 29 kg/m^2^ (26–35 kg/m^2^). The median duration of the ECMO therapy was 20 days (11–31 days), and the median ICU length of stay (LOS) was 35 days (22–57 days) [26]. The 102 patients accounted for a total of 2344 nutrition support days. Characteristics concerning the overall mean daily calorie and protein intake, as well as differences in baseline characteristics and nutrition support practices between the two subgroups (A and B) are depicted in Table 1.

### 3.2. Data on Position Dependent Nutrition Support

1830 of 2344 (78.1%) were spent in supine position and 514 (21.9%) in prone position. The position dependent energy and protein intake is shown in Figure 1, Table 2 and Appendix A. The mean daily energy intake was 74.4% (±29.6) on days in supine position and 71.1% (± 27.3) on days in prone position (*p* = 0.001). The proportion of enteral nutrition was significantly higher on days in supine position than on days in prone position (55.8% vs. 33.9%; *p* < 0.001). The opposite was found for parenteral nutrition (24.6% in prone vs. 11.2% in supine; *p* < 0.001). The mean daily protein intake was also significantly higher on the days in supine position compared to the days in prone position (0.7 vs. 0.68 g/kg BW/d; *p* = 0.007). In addition, significantly more propofol was administered on days in prone than on days in supine position (12.6% vs. 7.5% of calculated energy target; *p* < 0.001).

### 3.3. Nutrition Data in the First 30 Days of ECMO Support

No position-dependent difference in the mean daily energy intake was found in all the predefined time periods of the first 30 days of ECMO support (Table 2 and Appendix A). Regarding the type of nutritional intake (enteral vs. parenteral), it could be shown that on days 1–3, on days 1–7 and on days 15–30, the mean daily enteral energy and protein intake was significantly higher on days in supine position than on days in prone position. The lack of calories on days in prone position was compensated by a significantly increased administration of parenteral nutrition compared to the days in supine position (Table 2 and Appendix A).

### 3.4. Nutrition Support Practices during ECMO Therapy

Among the 2344 days with potential medical nutrition therapy, an adequate calorie intake was achieved on 952 days (40.6%; Table 3). The energy target was significantly more often reached on days in supine position than on days in prone position (42.1% vs. 35.4%; *p* = 0.007). On 956 days (40.8%), less than 70% of the calculated energy target was administrated. Undernutrition occurred significantly more frequently on days in prone position than on days in supine position (48.4% vs. 38.6%; *p* < 0.001). On 436 days (18.6%), the patients were overfed with no significant difference regarding their position. The observed significant differences in adequacy of nutrition supply between the days spent in prone versus supine position remained consistent upon analysis of the first 30 days of VV ECMO therapy (Table 3). On days in prone position, the proportion of days with only EN was significantly lower (48.1% vs. 69.3%; *p* < 0.001), and the proportion of days with tPN (16.3% vs. 5.3%; *p* < 0.001) or sPN (32.9% vs. 21.7%; *p* < 0.001) were significantly higher than on days in supine position (Table 3). EN was mainly delivered via nasogastric tube (2171 days, 92.6%).

There was no difference in the use of a post-pyloric tube or prokinetic therapy between days in supine versus prone position. In terms of metabolic complications associated with medical nutrition therapy, we reported at least 1 episode of hyperglycemia in 1306 (55.7%) nutrition support days, with a significantly higher occurrence on days in prone position (63.4% vs. 53.6%; *p* < 0.001). Hypertriglyceridemia was also significantly more common on nutrition support days in prone position (46.5% vs. 30.7%; *p* < 0.001, Table 3).

### 3.5. Data on GRV in Different Positions

The mean GRV was 111 (± 216.8) mL/24 h on days in supine position and 187.3 (±273) mL/24 h on days in prone position (*p* < 0.001; Table 4). A GRV ≥ 500 mL was documented on 196 of 2344 days (8.4%), with a significantly higher occurrence on days in prone position (15% vs. 6.5%; *p* < 0.001). In all the predefined time periods of the first 30 days of ECMO support, the mean GRV was significantly lower on days in supine position than on days in prone position (Table 4).

Moreover, we evaluated parameters associated with a GRV ≥ 500 mL/24 h by performing uni- and multivariable Cox regression analyses. As shown in Table 5, prone positioning not only increased the risk for a high GRV in univariable analysis (HR: 4.15 (CI 3.09–5.56), *p* < 0.001), it also proved to be independently associated with a high GRV in a multivariable Cox regression analysis (aHR: 4.06 (CI 3.00–5.50), *p* < 0.001) after adjusting for age (aHR: 0.99 (CI 0.98–1.01), *p* = 0.483), disease severity (SAPS II) at ECMO start (aHR: 0.98 (CI 0.96–1.00), *p* = 0.039), ECMO runtime (aHR: 0.95 (CI 0.93–0.96), *p* < 0.001) and ICU LOS (aHR: 1.00 (CI 1.00–1.01), *p* = 0.54).

### 3.6. Patient-Specific Tolerance of Medical Nutrition Therapy

In order to evaluate the patient-specific tolerance of enteral nutrition, we divided the cohort into two subgroups (A and B) according to their position during ECMO support (only supine versus supine and prone). One patient spent his whole ECMO runtime (seven days) in prone position and was therefore excluded from this calculation. Of the 101 patients, 28 (27.7%) patients were placed only in supine position (=subgroup A), while 73 (72.3%) patients alternated between supine and prone positioning during their ECMO support (=subgroup B). Patients in subgroup B had a significantly higher BMI (*p* = 0.017) as well as a significantly lower enteral (*p* < 0.001) and higher parenteral calorie delivery (*p* < 0.001, Table 1). No significant difference was found between the two subgroups regarding the overall mean GRV (*p* = 0.135, Table 6). The subgroup specific differences for daily calorie and protein delivery are shown in Appendix A. When comparing the patient-specific GRV in subgroup B in supine versus prone position, it was shown that the mean GRV on days in supine position was significantly lower than the mean GRV on days in prone position (*p* < 0.001, Table 6).

## 4. Discussion

In this retrospective study of 102 COVID-19 patients requiring VV ECMO therapy, we demonstrate that malnutrition is a common condition, and the tolerance of enteral nutrition depends on whether the patient is in prone or supine position. Prone positioning as a therapeutic approach in patients with moderate-to-severe ARDS improves oxygenation and has shown a survival benefit in several studies [30,31,32,33]. However, the available data regarding the tolerance of medical nutrition therapy in prone position is limited. Until now, it is largely unclear whether enteral nutrition in prone position leads to increased gastrointestinal dysfunction [34]. To the best of our knowledge, there are currently no data regarding the tolerance of enteral nutrition in prone position in COVID-19 patients requiring ECMO support.

In the present analysis, undernutrition, defined as calorie intake <70% of calculated energy requirements, was observed on 40.8% of the days, with a significantly higher occurrence on days in prone position. The mean daily calorie intake was significantly higher on days in supine position than on days in prone position. The mean daily protein delivery was 0.7 g/kg BW/d and therefore well below the recommendation of current guidelines [17]. When considering medical nutrition therapy in the defined time periods, the difference in position-dependent energy achievements could no longer be determined. This discrepancy could be explained as follows: after day 30 of ECMO therapy, the patients were predominantly placed in supine position as the respiratory situation, and thus the severity of critical illness, had already improved. Indeed, the time period beyond ECMO day 30 included 410 out of 2344 days of potential nutrition support days in our patient population. Of these, 389 (95%) were spent in supine position and 21 (5%) in prone position. As a result, the patients were able to receive more nutrition support in supine position from day 31 onwards, which had an impact on the overall mean daily calorie intake. In fact, the metabolism of critically ill patients is altered and medical nutrition therapy has to be adapted to the different metabolic phases [35,36]. Recent studies have shown that, contrary to previous assumptions, COVID-19 may not have a particular effect on the metabolic state of critically ill patients [37]. For this reason, the present study continues to be of notable relevance regarding the medical nutrition therapy of patients with ECMO support in the ICU.

GI dysfunction is common in critically ill patients and is characterized by functional impairment of the GI tract due to disturbances in motility and absorption, altered bile acid homeostasis, increased intra-abdominal pressure, changes in the microbiome, impaired mesenteric perfusion and local immune responses [12]. Therefore, a higher severity of critical illness is associated with a higher frequency of metabolic and gastrointestinal complications and, thus, reduced gastrointestinal tolerance [38,39]. Beside the critical illness itself, GI dysfunction is further exacerbated by the usage of high-dose sedation, neuromuscular blockade and opioids in ARDS patients [40]. In addition, hemodynamic instability, impaired microcirculation or intestinal barrier dysfunction due to the ECMO circulation may also negatively influence the tolerance of medical nutrition therapy [3,4,5]. However, data on nutrition support in ECMO patients is scarce. An increased GRV remains the main reason for the discontinuation of enteral nutrition [9,41]. Current ESPEN guidelines recommend a GRV of > 500 mL in 6 h to delay enteral feeding [17]. Prone positioning is no contraindication for early (within 48 h of ICU admission) enteral nutrition support [17]. However, our study showed that the mean GRV on days in prone position was significantly higher than the mean GRV on days in supine position, and a GRV ≥ 500 mL/24 h occurred significantly more often on days in prone position. Moreover, we demonstrated that prone positioning was independently associated with a high GRV after adjusting for age, disease severity (SAPS II) at the start of ECMO, ECMO runtime and ICU LOS. Therefore, a significantly higher amount of parenteral nutrition was administered on days in prone position to aim for adequate nutrition support. However, due to the increased parenteral nutrition on days in prone position, we reported a significantly higher incidence of metabolic complications, such as hypertriglyceridemia and hyperglycemia compared to days in supine position. Indeed, enteral nutrition shows a lower risk of overfeeding, more balanced nutrition intake, less infectious complications, less insulin resistance and shorter ICU and hospital stays, whereas mortality does not differ compared to parenteral nutrition [17,42,43]. Finally, we reported a significant increase in the administration of propofol on days in prone position, which also has a negative impact on GI function [38].

Currently available data on nutrition support practices in prone position remain controversial [23,24,34,44,45]. Van der Voort et al. [24] published a comparable median GRV in supine and prone position. Saez de la Fuente et al. [23] found no significant difference in GRV depending on the patient’s position. Savio et al. also reported no significant difference in enteral calorie intake between supine and prone position, meaning that in their study population, enteral nutrition was successful in prone position. Although the median GRV differed between supine and prone position, the amount of GRV showed no clinical relevance. In addition, there was no significant difference in the occurrence of a GRV > 250 mL [34]. All three studies mentioned above examined a patient cohort receiving only invasive mechanical ventilation without ECMO therapy [23,24,34]. A reason for the poorer tolerance of enteral nutrition in the present study could therefore be related to the severity of the disease and/or ECMO therapy and not to the prone position itself. Based on these data, it cannot be excluded that the inadequate nutrition support at least partially reflects the severity of the disease, which is known to be associated with a higher occurrence of metabolic and gastrointestinal complications [46,47]. Moreover, substantial evidence suggests that the COVID-19 infection induces gastrointestinal complications, such as high GRV, abdominal distension, vomiting, ileus and mesenteric ischemia [48,49,50], further complicating comparability.

In contrast, Alves de Paula et al. [45] and Alencar et al. [44] showed an influence of prone positioning on the tolerance of enteral nutrition in mechanically ventilated COVID-19 patients without ECMO support. Alves de Paula et al. documented a mean enteral nutrition at 70.0% of the calculated requirements in prone position and 74.8% in supine position [45]. In addition, there was an association between prone position and an increased GRV [45]. In the study by Alencar et al. [44], the mean daily enteral calorie intake was 88% of calculated requirements for patients maintained exclusively in supine position and 59% for those also placed in prone position.

There is currently no standardization of diagnostic or evidence-based therapeutic options for gastrointestinal dysfunction [12]. In our study, prokinetic drugs were administered on approximately half of the days, with no difference between days in prone and supine position. In our experience, prokinetic drugs were often used prophylactically and not only as a treatment for increased GRV. Therefore, no correlation can be made between the use of prokinetic drugs and gastrointestinal dysfunction. Overall, there was little use of post-pyloric tubes, probably due to the increased risk of bleeding events in anticoagulated ECMO patients. However, data on the use of post-pyloric tubes in ECMO patients are rare.

In addition to the position-dependent difference in the GRV, we report a mean GRV of only 128 mL in 24 h, which is well below the recommendation for delaying enteral nutrition in current ESPEN guidelines [17]. Despite this, patients were undernourished 40.8% of the days with potential nutrition support, indicating that enteral nutrition was probably stopped or reduced inadequately. In critically ill patients, medical nutrition therapy is often not the primary clinical focus, as priority is typically given to stabilizing hemodynamic and respiratory functions or managing other critical complications. This might explain why nutrition support, especially enteral, was compromised in the prone position. In order to preserve other vital functions in the patient, a reduction in nutritional intake may be considered an acceptable compromise. However, although nutrition might not be the primary factor in enhancing survival during the acute phase, it plays a critical role in long-term outcomes [26,51]. In addition, during the COVID-19 pandemic, patients infected with SARS-CoV-2 were isolated. In order to minimize patient contact for nursing staff and physicians, especially in the absence of vaccination, initiation/adjustment of medical nutrition therapy may have been given a lower priority.

## 5. Limitations

As this study represents a subanalysis, the data may overlap with the previous published study by Schneeweiss-Gleixner et al. [26]. Due to the retrospective character of the study, the data regarding the reasons for inadequate nutrition supply and interruptions of nutrition support are limited. In addition, daily energy requirements were calculated using simple weight-based equations, and the adequacy of nutrition support was defined according to current ESPEN guidelines, as indirect calorimetry is not feasible during ECMO support [17]. As a result, it is possible that energy targets were under- or overestimated in our analysis. Although there are experimental approaches for using indirect calorimetry during ECMO support in order to provide more accurate data, these protocols have not been evaluated for routine clinical use [52,53]. According to the Cox regression analysis, SAPS II at ECMO start and ECMO runtime seem to have a protective effect on the occurrence of a GRV ≥ 500 mL/24 h. Patients with the most severe illness (i.e., higher SAPS II and longer ECMO runtime) are likely to have received more parenteral nutrition than patients with a lower SAPS II and a shorter ECMO runtime, thus introducing a bias into this calculation. Positioning was categorized as prone whenever the patient had been in prone position for at least one hour on that day. An effort was made to maintain the prone position for at least 16 h according to current guidelines [29]. However, based on our definition of the prone position, it cannot be excluded that on some days categorized as prone position, a certain number of hours may have been spent in supine position.

Other limitations are the lack of a standardized definition of gastrointestinal dysfunction, the lack of adequate monitoring and the lack of evidence-based therapeutic approaches. Different publications used different definitions with different GRV thresholds and points of measurements. This situation is aggravated by the fact that the role of the GRV measurement as a predictor for gastrointestinal dysfunction remains unclear. The challenge in interpreting the results of this analysis is that, to the best of our knowledge, no study with a comparable patient cohort has been published so far. The present study examined the influence of prone positioning on the tolerance of medical nutrition therapy in critically ill COVID-19 patients requiring ECMO support. The literature search identified studies on nutrition support practices in ECMO patients without differentiation between prone and supine position (see Schneeweiss-Gleixner et al. [26]) and studies on the tolerance of enteral nutrition in prone position among mechanical ventilated patients [23,24,34,44,45]. However, there is no study that addresses both aspects within a single patient cohort. Therefore, the generalizability of our results is limited. Further, preferably prospective and interventional studies are needed to clarify the overall impact of prone positioning on medical nutrition therapy. These studies should be performed under dietary supervision and include comprehensive metabolic monitoring.

## 6. Conclusions

This data analysis presents nutrition-related data from 102 COVID-19 infected patients requiring VV ECMO support. We found that inadequate nutrition support is common in critically ill patients. Prone positioning was independently associated with a GRV ≥ 500 mL/24 h. Due to the retrospective study design, we cannot demonstrate causality, but our data suggest that prone positioning influences the tolerance of enteral nutrition support in COVID-19 patients requiring ECMO support. Future prospective studies are needed to confirm the influence of the patient’s position on medical nutrition therapy.

## Figures and Tables

**Figure 1 nutrients-16-03534-f001:**
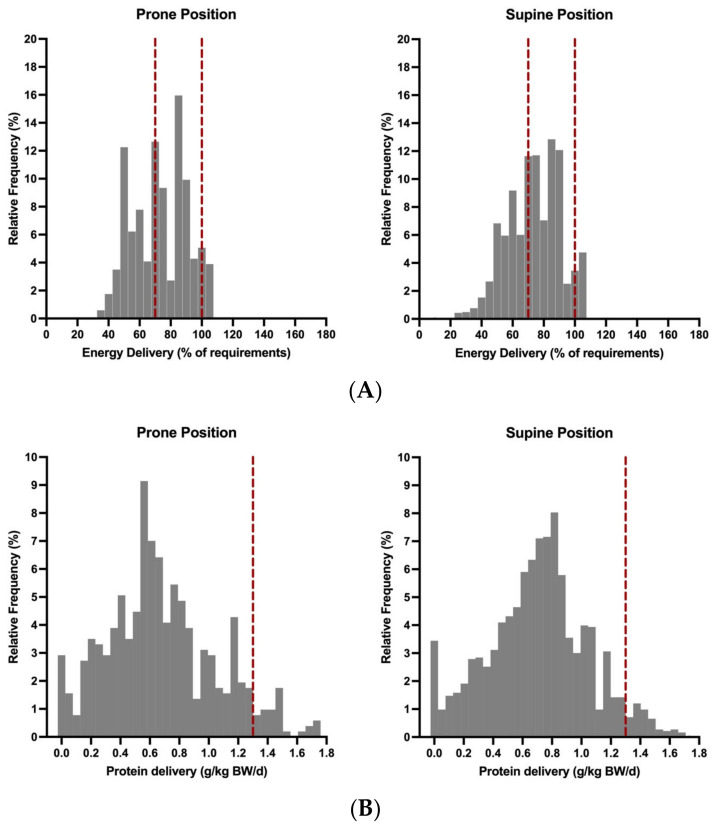
Distribution of energy (**A**) and protein (**B**) intake according to the patient’s position during ECMO support. Each value in this figure represents the nutrition delivery of one nutrition support day. For better comparability between the two groups, the relative frequencies are shown. The vertical red dotted lines in (**A**) indicate the values of adequate energy (70–100% of requirements) intake according to ESPEN guidelines. The vertical red dotted lines in (**B**) illustrates the ESPEN recommendations of 1.3 g/kg BW protein delivery per day.

**Table 1 nutrients-16-03534-t001:** Baseline characteristics according to patients only in supine position (subgroup A) and patients changing between supine and prone position (subgroup B).

Baseline Characteristics	Overall (n = 102 *)	Soubgroup A (n = 28 **)	Subgroup B (n = 73 **)	*p*-Value
** *Patient characteristics* **				
Male, n (%)	73 (71.6%)	23 (82.1%)	49 (67.1%)	0.135
Age (years), median (IQR)	57 (50–62)	57 (46–62)	57 (52–63)	0.719
Weight (kg), median (IQR)	90 (80–100)	85 (80–100)	80 (80–102)	0.447
BMI (kg/m^2^), median (IQR)	29 (26–35)	28 (26–31)	31 (27–35)	**0.017**
SOFA at admission, median (IQR)	8 (7–9)	7.5 (6.25–8)	8 (7–9)	0.313
SOFA at ECMO start, median (IQR)	8 (7–9)	8 (7–9)	8 (7–9.5)	0.695
SAPS II at admission, median (IQR)	42 (37–49)	40 (33.5–47.75)	42 (38–51)	0.183
SAPS II at ECMO start, median (IQR)	40 (34–46)	38.5 (33–44)	40 (35–48.5)	0.495
ECMO duration (days), median (IQR)	20 (11–31)	14 (9–28)	20 (12–31)	0.545
ICU LOS (days), median (IQR)	35 (22–57)	36 (24–53)	35 (20–58)	0.776
ICU mortality, n (%)	42 (41.2%)	10 (35.7%)	31 (42.5%)	0.536
** *Nutrition Data of 2344 days* **				
Daily calorie del. (% of requ.) overall, mean (Std.)	73.7 (29.1)	71.6 (26.0)	74.4 (30.0)	0.475
Daily calorie del. (% of requ.) from EN, mean (Std.)	51.0 (34.6)	56.2 (30.6)	49.4 (35.6)	**<0.001**
Daily calorie del. (% of requ.) from PN, mean (Std.)	14.2 (23.9)	9.4 (19.4)	15.7 (25.0)	**<0.001**
Daily calorie del. (% of requ.) from prop., mean (Std.)	8.6 (7.7)	6.0 (7.1)	9.4 (7.7)	**<0.001**
Daily protein del. (g/kg BW/d) overall, mean (Std.)	0.69 (0.35)	0.70 (0.32)	0.70 (0.36)	**0.037**
Daily protein del. (g/kg BW/d) from EN, mean (Std.)	0.49 (0.36)	0.55 (0.33)	0.47 (0.37)	**<0.001**
Daily protein del. (g/kg BW/d) from PN, mean (Std.)	0.21 (0.36)	0.15 (0.32)	0.23 (0.37)	**<0.001**
GRV (mL/24 h), mean ± Std.	127.8 (232.4)	107.6 (224.5)	134.5 (234.9)	**0.001**

* Data from Schneeweiß-Gleixner et al. ** Exclusion of one patient due to his position (prone only) during his ECMO support. Bold *p*-values indicate statistically significant results. Abbreviations: BMI—body mass index; BW—body weight; d—day; del.—delivery; ECMO—extracorporeal membrane oxygenation; EN—enteral nutrition; g—gram; ICU—intensive care unit; IQR—interquartile range; kg—kilogram; LOS—length of stay; m^2^—square meters; n—number; PN—parenteral nutrition; prop.—propofol; requ.—requirements; SAPS II—Simplified Acute Physiology Score II; SOFA—sequential organ failure assessment score; Std.—standard deviation; %—percent.

**Table 2 nutrients-16-03534-t002:** Data on mean daily energy delivery (% of calculated requirements).

Nutrition Data	Overall	Days in Supine	Days in Prone	*p*-Value
*All ECMO days*				
Overall, n (%)|mean (Std.)	2344 (100)|73.7 (29.1)	1830 (78.1)|74.4 (29.6)	514 (21.9)|71.1 (27.2)	**0.001**
EN, mean (Std.)	51.0 (34.6)	55.8 (34.3)	33.9 (29.9)	**<0.001**
PN, mean (Std.)	14.2 (23.9)	11.2 (21.2)	24.6 (29.4)	**<0.001**
Propofol, mean (Std.)	8.6 (7.7)	7.5 (7.4)	12.6 (7.3)	**<0.001**
** *ECMO days 1–3* **				
Overall, n (%)|mean (Std.)	305 (100)|63.9 (26.6)	183 (60)|61.8 (26.9)	122 (40)|67.1 (26.1)	0.086
EN, mean (Std.)	28.1 (27.2)	33.3 (28.7)	20.3 (22.7)	**<0.001**
PN, mean (Std.)	23.2 (27.0)	18.1 (23.9)	30.9 (29.7)	**<0.001**
Propofol, mean (Std.)	12.7 (6.7)	10.4 (6.4)	16.0 (5.7)	**<0.001**
** *ECMO days 1–7* **				
Overall, n (%)|mean (Std.)	693 (100)|67.3 (26.9)	433 (62.5)|66.1 (27.4)	260 (37.5)|69.2 (26)	0.137
EN, mean (Std.)	33.6 (29.9)	38.4 (30.9)	25.6 (26.2)	**<0.001**
PN, mean (Std.)	22.6 (27.3)	18.4 (25.6)	29.5 (28.5)	**<0.001**
Propofol, mean (Std.)	11.1 (7.1)	9.3 (6.7)	14.1 (6.6)	**<0.001**
** *ECMO days 8–14* **				
Overall, n (%)|mean (Std.)	548 (100)|75.1 (28.5)	428 (78.1)|74.4 (28.8)	120 (21.9)|77.3 (27.3)	0.464
EN, mean (Std.)	50.0 (34.5)	51.9 (35.1)	44.3 (31.9)	0.055
PN, mean (Std.)	17.2 (25.7)	16.0 (23.4)	21.6 (32.1)	0.344
Propofol, mean (Std.)	7.9 (7.2)	6.8 (6.9)	11.5 (7.4)	**<0.001**
** *ECMO days 15–30* **				
Overall, n (%)|mean (Std.)	693 (100)|74.1 (29.7)	580 (83.7)|74.8 (30.0)	113 (16.3)|70.6 (28.2)	0.072
EN, mean (Std.)	57.3 (32.2)	60.9 (31.5)	39.1 (30)	**<0.001**
PN, mean (Std.)	9.8 (20.4)	7.6 (17.8)	21.2 (27.9)	**<0.001**
Propofol, mean (Std.)	6.9 (7.4)	6.3 (7.2)	10.2 (7.7)	**<0.001**

Bold *p*-values indicate statistically significant results. Abbreviations: ECMO—extracorporeal membrane oxygenation; EN—enteral nutrition; n—number; PN—parenteral nutrition; Std.—standard deviation; %—percent.

**Table 3 nutrients-16-03534-t003:** Data on nutrition support practices.

	Overall	Days in Supine	Days in Prone	*p*-Value
*Energy delivery on all ECMO days*				
Total number of potential nutrition support days	2344	1830	514	
Days with calorie del. 70–100% of requ., n (%)	952 (40.6)	770 (42.1)	182 (35.4)	**0.007**
Days with calorie del. <70% of requ., n (%)	956 (40.8)	707 (38.6)	249 (48.4)	**<0.001**
Days with calorie del. >100% of requ., n (%)	436 (18.6)	353 (19.3)	83 (16.1)	0.106
** *Energy delivery on ECMO days 1–30* **				
Total number of potential nutrition support days	1934	1441	493	
Days with calorie del. 70–100% of requ., n (%)	769 (39.8)	597 (41.4)	172 (34.9)	**0.010**
Days with calorie del. <70% of requ., n (%)	852 (44.1)	613 (42.5)	239 (48.5)	**0.022**
Days with calorie del. >100% of requ., n (%)	313 (16.2)	231 (16.0)	82 (16.6)	0.754
** *Protein (g/kg BW/d) delivery on all ECMO days* **				
Days with protein del. ≥0.7 g/kg BW/d, n (%)	1187 (50.6)	972 (53.1)	215 (41.8)	**<0.001**
Days with protein del. ≥1.3 g/kg BW/d, n (%)	114 (4.9)	84 (4.6)	30 (5.8)	0.246
** *Protein (g/kg BW/d) delivery on ECMO days 1–30* **				
Days with protein del. ≥0.7 g/kg BW/d, n (%)	921 (47.6)	711 (49.3)	210 (42.6)	**0.010**
Days with protein del. ≥1.3 g/kg BW/d, n (%)	104 (5.4)	74 (5.1)	30 (6.1)	0.420
** *Nutrition support practices on all ECMO days* **				
Days with EN, n (%)	1516 (64.7)	1269 (69.3)	247 (48.1)	**<0.001**
Days with tPN, n (%)	181 (7.7)	97 (5.3)	84 (16.3)	**<0.001**
Days with sPN, n (%)	567 (24.1)	398 (21.7)	169 (32.9)	**<0.001**
Days with no nutrition support, n (%)	80 (3.4)	66 (3.6)	14 (2.7)	0.330
Days with prokinetic therapy, n (%)	1247 (53.2)	960 (52.5)	287 (55.8)	0.175
Days with post-pyloric tube, n (%)	151 (6.4)	113 (6.2)	38 (7.4)	0.320
Days with ≥1 episode of hyperglycemia, n (%)	1306 (55.7)	980 (53.6)	326 (63.4)	**<0.001**
Days with hypertriglyceridemia, n (%)	801 (34.2)	562 (30.7%)	239 (46.5)	**<0.001**

Bold *p*-values indicate statistically significant results. Abbreviations: BW—body weight; d—day; del. —delivery; ECMO—extracorporeal membrane oxygenation; EN—enteral nutrition; g—gram; kg—kilogram; n—number; PN—parenteral nutrition; requ.—requirements; sPN—supplemental parenteral nutrition; tPN—total parenteral nutrition; %—percent.

**Table 4 nutrients-16-03534-t004:** Data on GRV (mL/24 h).

Nutrition Data	Overall	Days in Supine	Days in Prone	*p*-Value
*Gastric residual Volume (GRV)*				
All ECMO days, n (%)|mean (Std.)	2344 (100)|127.8 (232.4)	1830 (78.1)|111.0 (216.8)	514 (21.9)|187.3 (273.0)	**<0.001**
ECMO days 1–3, n (%)|mean (Std.)	305 (100)|147.3 (218.2)	183 (60)|117.1 (188.4)	122 (40.0)|192.7 (250.5)	**0.002**
ECMO days 1–7, n (%)|mean (Std.)	693 (100)|157.3 (247.2)	433 (62.5)|139.5 (239.4)	260 (37.5)|186.9 (257.6)	**<0.001**
ECMO days 8–14, n (%)|mean (Std.)	548 (100)|143.1 (224.5)	428 (78.1)|134.2 (214.2)	120 (21.9)|175 (256.5)	0.131
ECMO days 15–30, n (%)|mean (Std.)	693 (100)|122.5 (245.7)	580 (83.7)|104.3 (226.9)	113 (16.3)|215.9 (310.6)	**<0.001**
GRV ≥ 500 mL on all ECMO days, n (%)	196 (8.4)	119 (6.5)	77 (15.0)	**<0.001**
GRV ≥ 500 mL on ECMO days 1–3, n (%)	28 (9.2)	10 (5.5)	18 (14.8)	**0.006**
GRV ≥ 500 mL on ECMO days 1–7, n (%)	72 (10.4)	37 (8.6)	35 (13.5)	**0.04**
GRV ≥ 500 mL on ECMO days 8–14, n (%)	53 (9.7)	35 (8.2)	18 (15)	**0.025**
GRV ≥ 500 mL on ECMO days 15–30, n (%)	55 (7.9)	32 (5.5)	23 (20.4)	**<0.001**

Bold *p*-values indicate statistically significant results. Abbreviations: ECMO—extracorporeal membrane oxygenation; GRV—gastric residual volume; h—hour; mL—milliliter; n—number; Std.—standard deviation; %—percent.

**Table 5 nutrients-16-03534-t005:** Univariable and multivariable analyses of prognostic factors for a GRV ≥ 500 mL/24 h.

Parameter of Interest	Univariate (Unadjusted) Analysis			Multivariate (Adjusted) Analysis		
GRV ≥ 500 mL/24 h	HR	95%CI	*p*-Value	aHR	95%CI	*p*-Value
Prone positioning, yes	4.15	3.09–5.56	**<0.001**	4.06	3.00–5.50	**<0.001**
Age (years)	0.97	0.95–0.98	**<0.001**	0.99	0.98–1.01	0.483
Sex (male)	1.10	0.79–1.54	0.584	-	-	-
BMI (kg/m^2^)	1.02	1.00–1.05	0.097	-	-	-
SAPSII at ECMO start	0.99	0.97–0.99	**0.04**	0.98	0.96–1.00	**0.039**
ECMO runtime	0.95	0.94–0.96	**<0.001**	0.95	0.93–0.96	**<0.001**
ICU LOS	0.908	0.97–0.99	**0.001**	1.00	1.00–1.01	0.54

Bold *p*-values indicate statistically significant results. Abbreviations: aHR—adjusted hazard ratio; BMI—body mass index; CI—confidence interval; ECMO—extracorporeal membrane oxygenation; GRV—gastric residual volume; HR—hazard ratio; h—hour; ICU—intensive care unit; kg—kilogram; LOS—length of stay; m^2^—square meters; mL—milliliter; SAPSII—Simplified Acute Physiology Score II; %—percent.

**Table 6 nutrients-16-03534-t006:** Patient-specific tolerance of medical nutrition therapy.

	Mean GRV (mL/24 h) Overall, Mean (Std.)	Mean GRV (mL/24 h) in Supine, Mean (Std.)	Mean GRV (mL/24 h) in Prone, Mean (Std.)	*p*-Value
*Group A*	101.4 (92.9) *	101.4 (92.9) **	/	/
*Group B*	146.5 (106.7) *	130.1 (138.5) **	199.1 (175.7)	**<0.001**

* *p*-value for the overall mean GRV compared in group A vs. group B; *p* = 0.135. ** *p*-value for mean GRV in supine position compared in group A vs. group B; *p* = 0.294. Bold *p*-values indicate statistically significant results. Abbreviations: GRV—gastric residual volume; h—hour; mL—milliliter; Std.—standard deviation.

## Data Availability

The data is available upon reasonable request to the corresponding author.

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
