# Peer review of "The Influence of Prone Positioning on Energy and Protein Delivery in COVID-19 Patients Requiring ECMO Support"

_nutrients, 2024, doi:10.3390/nu16203534_

Round 1
Reviewer 1 Report
Comments and Suggestions for Authors
This study is very great for readers. However, several concerns must be revised.
Abstract:
To reformulate the background/aim: "Gastrointestinal dysfunction is a common complication of medical nutrition therapy in critically ill patients. Whether prone positioning leads to a deterioration in GI function has not been fully clarified." but what is aim?
In addition, to add more numbers (values) in the results section, the p value few show/explain the data.
Introduction:
What is hypothesis?
Methods:
What are references to do SOFA and food consumption?
Results:
What are food intake? calories? carbo? protein? lip? in g and calories?
What are presence of chronic diseases and medicines use? Enteral nutritional suffer alteration with use of medicines.
Are anaemic patients? Show the data and discuss it.
Discussion:
Lack findings regarding to microbiota. What is impact of enteral nutrition with omega 3 x without omega 3.
Reviewer 2 Report
Comments and Suggestions for Authors
The manuscript focuses on the adequacy of energy and protein intake in COVID-19 patients undergoing VV ECMO, with a focus on the correlation between insufficient caloric intake and increased mortality in the ICU. The authors used a cohort of 102 patients, with data collected retrospectively. Although the topic is of great clinical relevance, there are some critical issues that should be addressed to improve the quality of the work.
1. The multidisciplinary orientation of ECMO placement:
I believe that the decision to place patients in the prone position, especially in an ECMO setting, should be evaluated by a multidisciplinary team. This decision takes into account various factors, including improved oxygenation, risk of lung damage and other complications. Nutrition, although important, is not always the main priority in these circumstances. It would be helpful for the authors to better contextualise the role of nutrition in relation to other clinical priorities in such a critical situation.
2. Secondary role of nutrition in the prone position:
In emergency situations, such as the management of patients with severe ARDS and COVID-19, it is understandable that nutrition may take second place to respiratory stabilisation. This might explain why nutritional support, especially enteral, was compromised in prone patients. The authors should make it more explicit that, in some cases, reducing nutritional intake may be an acceptable compromise to maintain other vital functions of the patient.
3. Considerations on prospective studies:
Although the study was conducted in a rigorous manner, its retrospective nature limits causal conclusions. The authors might suggest the need for prospective studies that more accurately assess the overall impact of pronation positioning on nutrition. It would be particularly interesting to include multidisciplinary evaluations that balance the importance of nutrition against other critical patient functions.
4. Methodological criticisms:
A major limitation of the study is the use of body weight-based estimates for energy requirements, which may not accurately reflect the true metabolic needs of patients undergoing ECMO. The use of indirect calorimetry, where possible, could provide more accurate data. The authors should consider this limitation and discuss its impact on the results.
5. Lack of a recorded protocol:
It would be good if the authors had followed a registered protocol to improve methodological transparency and reduce the risk of ‘salami-slicing’, as the data seem to overlap with other publications.
In conclusion, the manuscript covers a topic relevant to the management of critically ill patients, but needs more attention to the multidisciplinary context and clinical priorities. I recommend a review to better clarify the role of nutrition in overall ECMO treatment, while suggesting prospective studies to confirm the findings.
Round 2
Reviewer 1 Report
Comments and Suggestions for Authors
Accept
Reviewer 2 Report
Comments and Suggestions for Authors
The authors have addressed all my cornerns.